# Quaternary Glauconitization on Gulf of Guinea, Glauconite Factory: Overview of and New Data on Tropical Atlantic Continental Shelves and Deep Slopes

**Pierre Giresse**

Centre of Education and Research on Mediterranean Environments (CEFREM), UMR CNRS 5110, Perpignan Via Domitia University, 66860 Perpignan, France; giresse@univ-perp.fr

**Abstract:** For a long time, particular attention was paid to glauconitization in the surficial sediments lying on the outer continental shelves of present oceans. Subsequently, the processes observed and analyzed may have served as models for studies of glauconite in Cenozoic or even Mesozoic shelf deposits. Access to the sedimentary domains of deep oceans, particularly those of contouritic accumulation fields, has made it possible to discover unexpected processes of glauconitization. Thus, the long-term prevalence of control using fairly high-temperature water has become obsolete, and the prerequisite influence of continental flows has come to be considered on a new scale. Frequently, sediments from contouritic accumulation provide a condensed and undisturbed sedimentary record without periods of sediment erosion. Glauconitic grains could possibly integrate the signatures of bottom-water masses over prolonged periods of time, which, while preventing their use in high-resolution studies, would provide an effective means of yielding reliable average estimates on past $\varepsilon Nd$ signatures of bottom-water masses. In this regard, glauconitic grains are probably better-suited to paleoceanographic reconstructions than foraminifera and leached Fe-oxyhydroxide fractions, which appear to be influenced by sediment redistribution and the presence of terrestrial continental Fe-oxides, respectively. Direct methodological access to the compositions of the semi-confined microenvironments of neoformation has largely renewed the information, chemical or crystallographic, that was previously, and for a long time, restricted to macromeasurements. The various granular supports (mudclasts, fecal pellets, and foraminifera infillings) include inherited 1:1 clays (or Te-Oc; i.e., clay minerals consisting of one tetrahedral sheet and one octahedral sheet, such as kaolinite) that are gradually replaced by 2:1 clays (Te-Oc-Te) dominated first by smectite, and then by glauconite. In small pores, the water's activity is diminished; as a consequence, the precipitation of a great number of mineral species is thereby made easier, and their stability domains are changed. A specific methodological approach allows the study of the mineralogy and chemistry of the fine-scale mineral phases and to avoid the global aspect of the analytical methods previously used in the initial studies. Wide-field micrographs taken at a mean direct magnification of 100.000 show the intimate and characteristic organization of the main phases that occur in a single grain. One or several "fine" (about 10 nanometers in scale) microchemical analyses can be recorded, and directly coupled with each interesting and well-identified structure image observed in HRTEM.

**Keywords:** glauconite; shelf; contouritic slopes; winnowing; iron; Gulf of Guinea; Ivory Coast–Ghana Ridge; Demerara margin

## 1. Introduction

This study does not propose a synthetic work considering the exhaustive state of our knowledge on glauconitization through the oceans of the world. It aims even less at dealing with the fossil glauconite discovered in Cenozoic or even older terrains. Its primary objective is to present a review of knowledge on the recent processes of glauconitization in the southern Gulf of Guinea (Congo, Gabon, Cameroon, and Ghana, in particular). This

sector is undoubtedly one of the most efficient ocean "factories" of glauconite and, as such, should soon be the target of new and important research programs. The data that I present here are the result of a scientific journey of some 30 to 40 years, in which I have had several opportunities to participate. In addition to several fairly recent articles, some older articles are little known, partly because they are written in French and published in journals with limited circulation. This present article is an opportunity to make them public after having revisited them and extracted the most essential points in light of recent scientific progress.

The authigenic clay mineral, glauconite, grows at the sediment–water interface (henceforth called glauconitization) during the early marine diagenesis of Quaternary deposits. It is generally agreed that the degree to which green grains have become mineralogical glauconite (a 10 Å mineral) is an indicator of the length of the depositional hiatus that promoted its process [1–5]. The residence time is expressed by the increasing concentration of potassium [3,4,6].

On the outer edges of shelves, the formation of glauconitic grains in shallow-marine sediment corresponds to a condensed horizon developed over a period of time, which was controlled by the last glacio-eustatic oscillation. One important condition for this formation to occur is a sufficiently long exposure time of the grains at the water–sediment interface, allowing the capture of iron and potassium in the newly formed Te-Oc-Te structures. Under these general conditions of low sediment accumulation, only one glauconitic sequence can be deposited, but it is fated to be destroyed during the low sea-level that follows. Autochtonous glauconite is common in the Transgressive Systems Tract (TST) and the Lower Highstand Systems Tract (LHST), indicating an upward increase followed by a decrease in both concentration and mineralogical maturity. Consequently, a pluri-sequential accumulation at the Pleistocene scale is not registered on the shelf.

On deeper oceanic slopes (more than 2000 m in water depth) swept by powerful contour currents, the bottoms are the object of a winnowing, which, by almost permanent reworking, delays the burial and controls through a prolonged residence at the water–sediment interface of the glauconitic grains, allowing the progress of their mineralogical maturation. However, these winnowing conditions can sometimes induce more or less marked stratigraphic gaps, which disrupt stratigraphic analysis. Despite their sometimes incomplete character on the scale of the last isotopic stages, the contouritic accumulations offer a glauconitic archive, often allowing the geochemical analysis of paleooceanic changes [7]. The vertical fluctuations in green-grain concentrations correspond to microsequences that are linked to the pulses of energy of the bottom water.

Various types of substrate serve as supports for glauconite growth in contouritic deposits, but foraminifera and other microfossil fillings are the most common. These initial substrates are distinct from the clay fecal pellets observed on various continental shelves, such as those of the Gulf of Guinea.

The temperatures reported for glauconitization from low-latitude sea bottoms of the continental shelf are generally about or slightly below 15 °C. For the first time, glauconitization proceedings were documented near 3 °C at water depths beyond 2100 m on the Ivory Coast–Ghana Ridge and Guyana margins [7,8].

Iron-rich terrigenous supply from a nearby continent is considered as a prerequisite for the glauconitization process. From this point of view, ocean bottoms off the intertropical continents subjected to intense hydrolysis processes are favored. However, glauconitization can progress under the same rhythms on the underlying oceanic slope, despite their greater distance from the iron source [5,7,8].

## 2. Marine Muds as Sites of Glauconitization

In their synthesis of the geology of the Atlantic Ocean, Emery and Uchupi [9] wrote: "The mineral glauconite consists of sand-sized grains very widespread on continental shelves, bank tops, and upper slopes, extending rarely and locally to water depths of 1000 m". Effectively, for a long time, attention has been paid to the unburied sediments lying on the continental shelves of present oceans.

### 2.1. Megascale Approach

The general concept of a largely ubiquitous glauconitization process that affects the ocean floors of the globe is acceptable because it is difficult to find marine muds that are completely devoid of glauconitic green grains. Thus, around France, it is possible to observe recent shelf sediments, such as those of the Mediterranean [10] or the English Channel [11], that contain very small number of whitish or pale-green grains, that is to say, that are poorly evolved. Green clay grains are largely observed through the entire shelf and upper slope of the Atlantic margin. On the other hand, if we consider, in a restrictive way, oceanic sediments containing more than 10 wt.% of glauconitic grains, we define a distribution area that is restricted to intertropical latitudes and, in particular, off the mouths of large rivers flowing into the African margin, such as the Congo, the Ogooué, and the Niger, to name only the most significant. This distribution is illustrated by the total-iron-contents map of the western margin of Africa by Emelyanov [12] (Figure 1). The same observation can be verified on the South American margin, particularly off the Orinoco and Amazon mouths; the latter river constitutes a special case insofar as the powerful flow and the ocean currents disperse alluvial contributions at a very great distance from the river mouth. In recent decades, the discovery of glauconitic concentrations in contourites on the slopes of the Ivory Coast–Ghana Ridge [13,14] and at the Demerara margin in Guyana [8,15] has made it possible to further extend this general concept of glauconitogenic muds developing in intertropical oceanic zones.

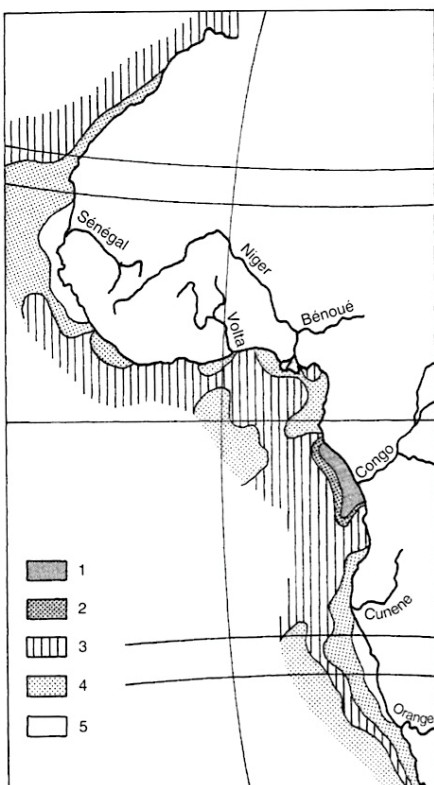

**Figure 1.** Iron deposition in the upper 0.5 cm of the sediments of the Atlantic margin of Africa. Iron contents ($Fe_2O_3$ wt.%) were measured on the terrigenous fraction (without carbonate, organic carbon, or biogenic silica); 1. >10%, 2. 7%–10%, 3. 5%–7%, 4. 3%–5%, 5. <3%, modified after Emelyanov [12].

The distributions of two of the components of glauconitic muds need to be analyzed with particular attention. These components are iron and organic matter, the latter expressed in the form of C org. These two components are relatively well documented in the context of the Gulf of Guinea.

## 2.2. *Example of Organic and Ferruginous Accumulations off the Mouth of the Congo River*

The Cabinda and southern Congo shelves are the sites of the accumulation of part of the suspended matter from the Congo River, the other part being deeper offshore [16]. The turbid waters brought by the NW current of the river first begin to settle on the outer edge of the shelf, and then move towards the coast (Figure 2a). This defines an extensive area of high accumulation rates, where fecal pellets are poorly concentrated. To the north of Pointe-Noire, the inflow from the river weakens and, consequently, evidence of Holocene transgressive deposits often emerges: around 100 m, there are mostly coastal shelly sands and, around 110–120 m, glauconitic green sands.

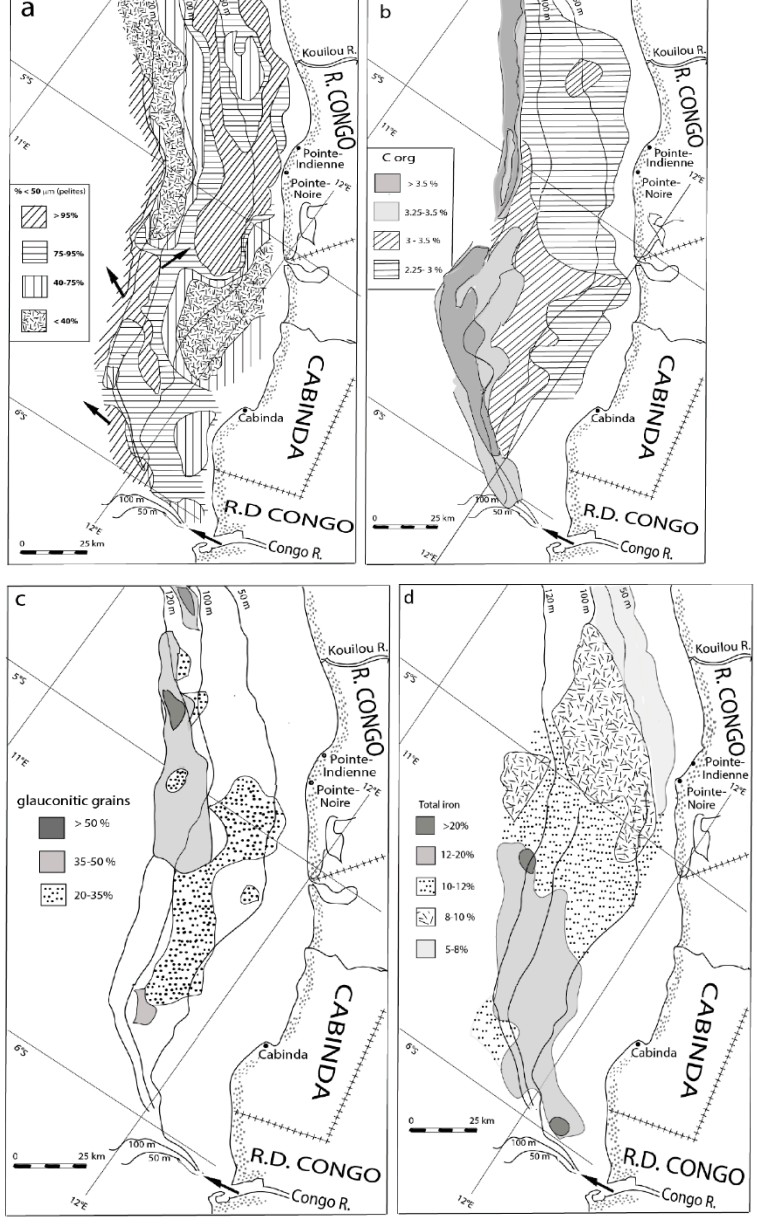

**Figure 2. (a)**. Distribution map of the concentrations of pelitic matter (<50 µm): >95%, 75%–95%, 40%–75%, 15%–40%, less than 15%. **(b)** Distribution map of the concentrations of total C org: >3.5%, 3.25%–3.5%, 3%–3.25%, 2.25%–3% (redrawn from Giresse and Moguedet, [16]. **(c)** Distribution map of the high concentrations of glauconitic grains related to the sandy fraction of the sediment: >50%, 35%–50%, 20%–35%. **(d)** Distribution map of the total iron in the silty muds >20%, 12%–20%; 10%–12%; 8%–10%; 5%–8% (redrawn from Giresse, [1,2].

The C org contents of the fraction of the muds below 50 μm show highest concentrations (>3.5 wt.%) on the outer edge of the shelf off the mouth of the Congo ([16], ibid). Subsequently, they decrease slightly off Pointe-Noire, 50 km further north (Figure 2b).

The highest glauconitic concentrations are mainly located on the outer edge of the D.R. Congo shelf, with maxima exceeding 50 wt.% of the sandy fraction [1,2]. On the other hand, glauconitization is also observed on the outer edge of the Cabinda plateau, where the rate of alluvial mud deposition remains high. Finally, a S-N band with a moderate concentration (25 to 35 wt.%) extends diagonally from the Cabinda offshore to Pointe-Noire; it corresponds to a decreasing deposition of suspended matter when moving away from the mouth of the Congo (Figure 2c).

Measurements of pH were taken directly inside the wet sediment ([16], ibid). The alluvial muds of the outer edge of the Cabinda plateau, with a pH average of less than 7.8, are the most acidic. Further north, away from areas of active sedimentation, the acidity decreases more or less regularly to reach pH values ranging between 8 and 8.2. Finally, in the green sands of the outer shelf, the pH is close to or greater than 8.2, despite the association with a still significant pelitic and organic fraction. We thus observe a distribution of pH taking shape that broadly matches that of the organic carbon. This parameter of the sedimentary matrix of pellets undergoing glauconitization is likely to participate in the control of cationic exchanges; unfortunately, it is not possible to measure the pH within the microenvironments of the pellets [17].

The significance of the abundant supply of total iron at shallow depths (20–60 m) near a tropical river mouth was emphasized as one major factor in the first steps of glauconitization [18]. This proximity of the terrigenous source was associated with a relatively active accumulation rate, which controlled a restricted period of ion exchange at the sediment–water interface. These nascent or slightly evolved grains therefore seem linked to relatively important river supply, but not necessarily to very shallow depths. According to the importance of the river flux and to the current interaction, the mud deposits of the mid-shelf currently seem to be a very favorable setting. However, in the case of very strong river input, the most favorable sedimentary environment for glauconitization is shifted to the outer shelf or the upper slope. This is illustrated by the Cabinda shelf example off the Congo River deposition [18], and by the outer shelf of Cameroon, where a part of the Niger River supply is presently accumulating [19].

Significant total iron contents are measured in the mud near the mouth of the Congo River (26.3, 21.2 and 14.3 wt.%). Moving further away, we find lower and more irregular contents (6.1 to 1.4 wt.%), mainly related to the coarser texture of the sediment (Figure 2d). These contents do not necessarily signify the progress of glauconitization because, as shown below, part of this iron is rapidly trapped within the pellets in the state of oxyhydroxides.

The most frequent support of glauconitization is that of the fecal pellets of Polychaetes, which mainly feed on the bacteria that they ingest, consuming about half of their cells [20]. Subsequently, important biochemical energy can be supplied during the various stages of neoformation. The Congo River is likely to feature inoculum of continental origin, as in the case of the ferro-manganic grains observed off the mouths of the Wouri, on the Cameroonian shelf [21,22]. This type of biomediation has not been clearly demonstrated in Congo pellets; however, it is illustrated and strongly suggested in the grains undergoing glauconitization from the contouritic deposits of the Ivory Coast–Ghana Rigge [14].

## 3. Fecal Pellet Glauconitization on the Gulf of Guinea Shelf

On the Gulf of Guinea shelf *sensu lato*, the ellipsoidal fecal pellets of mud-eating Polychaetes represent the largely prevailing substrate and are suggestive of a muddy initial sedimentary settling.

However, on the outer edge, in the sandier accumulations deposited during the beginning of the last sea-level rise, shell debris or foraminifera infillings provide the most frequent supports for glauconitization.

Previously, my attention was drawn by the prevalent role of the semi-confined microenvironment that these granular supports can play with respect to the ambient water environment of the sediment [3]. Cationic exchanges with the surrounding interstitial water initiate various steps of crystallization, at first concerning only a restricted part of the grain. A two-step process was also suggested for Jurassic and Cretaceous carbonate rocks from SE Spain. Most peloids are K rich and show well defined 10 Å lattice fringes, whereas the shielded microenvironments of other smectitic peloids with lower K contents indicate a glauconitic precursor [23]. The function of this microenvironment is controlled by its geometry: if it is too small (i.e., less than 100 μm), it is also permeable, the ion departures are preponderant on the entrapment and, subsequently, the support is intended to be quickly mechanically dispersed. If it is too large (approximately more than one mm), the cationic exchanges cannot reach the innermost (or the deepest) parts, and only the most proximal zones become the simultaneous sites of the fixations and leachings that condition the neoformation process. As a result, the glauconitization will mainly concern supports of intermediate size, around 200 to 500 μm in diameter (Figure 3).

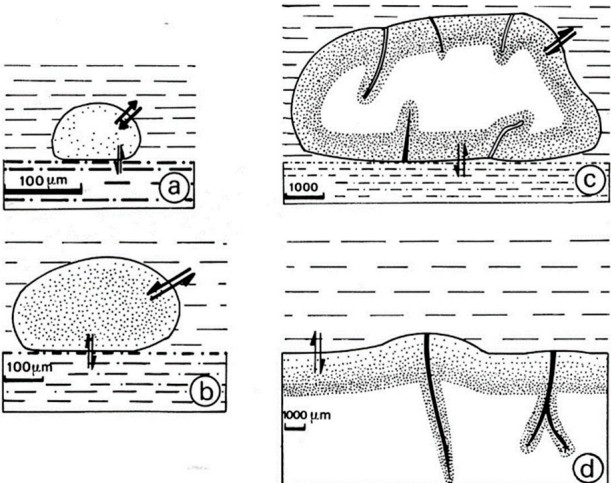

**Figure 3.** Glauconitization of substrates differing by size at the seawater (or interstitial water)–fecal pellet interface. (**a**) Small grains; (**b**) coarse grains; (**c**) lithoclasts; (**d**) hard ground. Small dots indicate the density of the glauconitization process and arrows represent cation exchange, after Odin and Matter [3].

This geometric scale of cation exchanges was the subject of a scheme applied to a fecal pellet measuring one mm in diameter [23]. Subsequently, we can distinguish (1) an external and porous transit sector, where the inherited mineral phases dominate the newly formed phases and where the first precipitations of iron oxyhydroxides are rapidly defined; (2) a deeper middle sector, where both the departures and arrivals of the cations are efficient, which is the privileged site of the neoformations of glauconitization; and (3) a more internal sector, which remains free of glauconitization (Figure 4). In the vast majority of the shelves in the Gulf of Guinea and, in particular, on the Congolese and Gabonese shelves, the optimum diameters of fecal pellets remain below 500 μm and, therefore, do not allow the glauconitization of the innermost sector. In the 300–400-micrometer Congolese pellets, it is always the innermost part that shelters the sites where glauconitization is the most advanced [7,23].

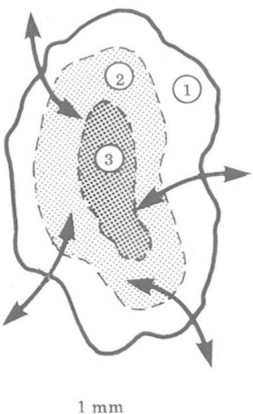

**Figure 4.** Zonation in a 1-millimeter-wide ellipsoidal fecal pellet. 1: transit sector; 2: mineralogenesis sector; 3: sector isolated from surrounding environment, after Odin [23].

To this day, the lines of G. Millot's introduction to the book of Odin [4] are still relevant: "We understand that the micromilieu found at the center of the grain or pellets presents physicochemical and biogeochemical characteristics different from those of seawater where crabs and turbots swim. Now this is of great consequence. For most available thermodynamics models provide for mineral-solution conditions in equilibrium, in diluted solution, where the activity of the water is taken as unity".

### 4. Timing of Glauconitization on the Gulf of Guinea

The majority of the Atlantic shelves of the intertropical African continent are sites of significant glauconitization processes that affect Holocene mud areas and, to an even greater extent, the relict deposits contemporary with or slightly posterior to the last drop in sea levels.

Here, the Congolese shelf is related to the area of the Congo River that flows into the Atlantic. In fact, it successively includes the R.D. Congo (formerly Zaire) shelf, the Cabinda shelf, the R. Congo shelf, and a part of the southern Gabon shelf. The Gabon and Congo (R. Congo) shelves are where the succession of late-Quaternary deposits is the most completely documented and, in particular, the best dated [1,2] (Figure 5). Schematically, we distinguish two distinct environments:

- Accumulations of sandy mud on the middle and inner shelves extending off the mouths of large rivers (Congo, Ogooué, Niger), which are dated to the Holocene (here post-7,000 years BP). These are muds with fairly high Holocene sediment-accumulation rates (ranging from 4 to 10 g cm$^{-2}$ 10$^{-3}$ years), which, locally, can reach thicknesses of up to ten meters [24]. These organic-rich muds are sites of benthic life, in particular that of limivorous polychaete worms, whose fecal pellets in these areas reach only the beginning stage of greening. Their mineralogical composition is still often dominated by the Fe-kaolinite minerals inherited from the continent; however, the first neoformations of Fe-Mg TeOc minerals are observed, as well as the precipitation of aggregates of iron oxyhydroxides or, sporadically, manganese which can extend as far as the formation of pseudo-ooids. These oxyhydroxides relate to various mineral impurities rather than glauconite structures. At this stage of weak evolution, the neoformation of TeOcTe phases is still at its beginning. The contents of green grain sediments, below 5 wt.% near the coast, increase steadily away from the coast, reaching almost 25 wt.% towards water depths of 90 to 100 m, where the deposits are slightly older [25].
- Relict remains of the last Pleistocene low sea level emerge at depths between 110 m at the top of the slope and near 130 m. Around 110 m in water depth, there are coastal shell sands with abundant Amphistegine shells that have been repeatedly dated between 12,000 and 10,000 years BP; numerous limestone clasts in the process of glauconitization are added to the fecal pellets to reach green-grain contents between

20 and 30 wt.%. At depths between 120 and 130 m, the glauconitic contents rise to more than 80 wt.%, constituting true glauconitite deposits. These deposits testify to repeated episodes of reworking, involving chemical exchanges at the water–sediment interface that are probably still ongoing. These fecal pellets are cracked and dark green. The most advanced stages of glauconitization are found with neoformed Te-Oc-Te sheets of the K-nontronite type or K-beideillite type, but TeOc structures at 7 Å are still present [18,25]. The accumulation of these green grains can be considered as postdating marine or lagoon deposits attributed globally to stage 3 (>35,000 years BP, according to radiocarbon measurements), whose strong compaction testifies to the emergence during the last low sea level. Subsequently, these green grains began their glauconitization process during the short course of stage 2; it is not possible to specify further. In deeper waters, on the tops of slopes and without emersion episodes, the muds were sites of continuous glauconitization, albeit with weak development due to regular and fairly rapid burial limiting the lengths of cationic exchanges.

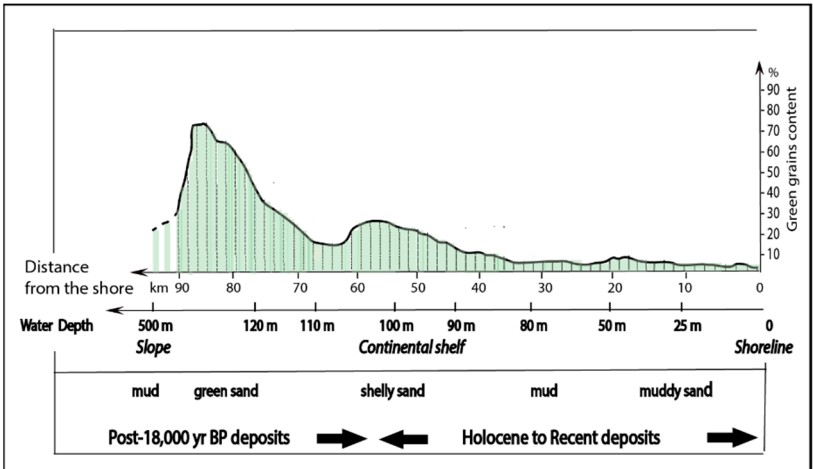

**Figure 5.** Synthetic section showing, from the coast out to sea, the distribution of sediments. Glauconitic grains are especially abundant on the outer part of the shelf and the top of the slope (redrawn from [26–28] for the chronology and from [29] for the mineralogy.

In summary, one group of grains is characterized by high concentrations of terrigeneous iron and a slight increase in marine magnesium and potassium: this slightly evolved mineralogical composition corresponds generally to young deposits of the inner part of the shelf or to ancient deposits that were rapidly buried. A second group is characterized by a consequent increase in marine potassium and, to a lesser extent, in marine magnesium, and corresponds to older deposits in the outer shelf.

## 5. Sequential Approach to Glauconitic Accumulation on the Congo Shelf

On the shelf off the mouth of the Congo River, the definition of a glauconitic sequence is restricted to the palaeoenvironmental evidence of the last glacio-eustatic oscillation. According to this oscillation, which was similar along the entire West African margin [26–28], a single glauconitic sequence, immediately posterior to the last low-level stand, was preserved above bottom depths of 120 m. The first steps in this sequence can thus be dated to approximately 20,000 years BP. It is a transgressive deposit which, in the middle part of the shelf, is discordant above the alluvial or marshy deposits contemporaneous with the last emersion, while in the outer part, it covers infralittoral or circalittoral marine muds that were frequently compacted and planed during the subsequent low sea level (see next § 6) (Figures 6 and 7).

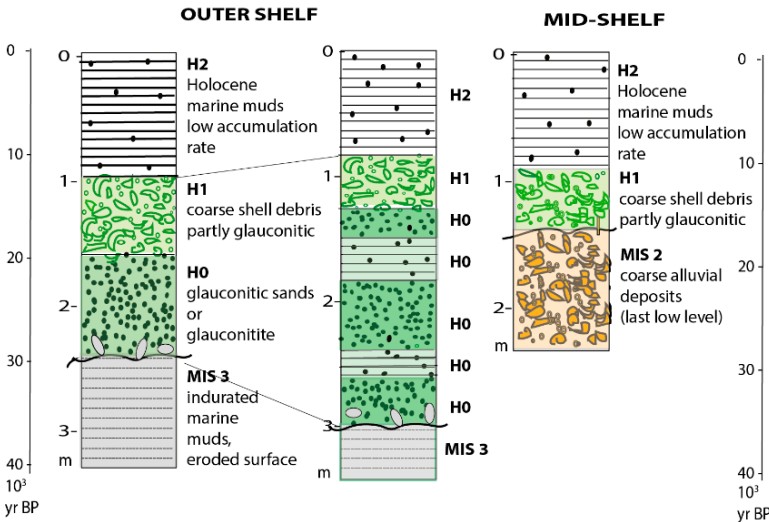

**Figure 6.** Typical sequences observed on the outer edge and the middle part of the plateau. MIS 3: infralittoral indurated muds. H0: glauconitic sands from the last low sea-level and from the start of the Holocene transgression. H1: shell deposits from the beginning of the Holocene sea-level rise. H2: muds from the end of the rise and from the last high sea level, redrawn from [27].

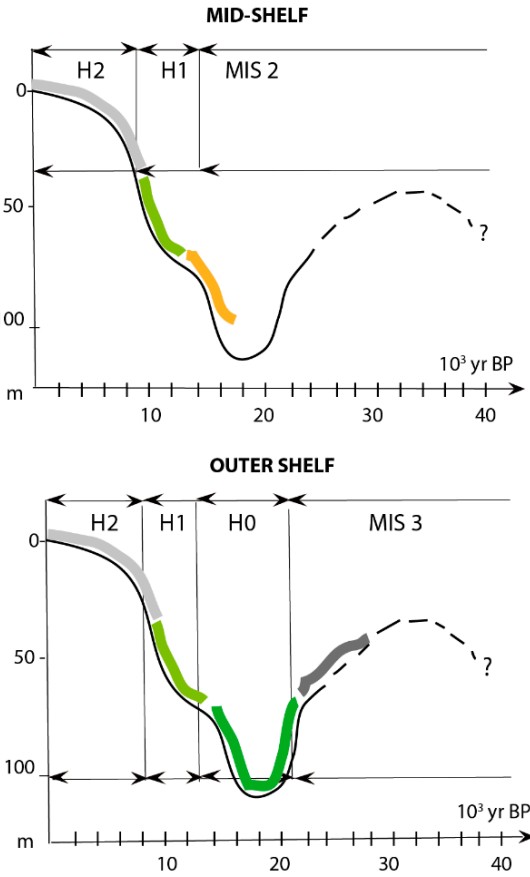

**Figure 7.** The main sedimentary sequences on the mid-shelf and the outer shelf of the Congolese plateau, according to the glacio-eustatic movement of the shoreline. H2: late Holocene (muddy near the mouth of the Congo). H1: shell barrier from the beginning of the Holocene sea-level rise. H0: pre-Holocene regression glauconitic sands. MIS2: alluvial deposits of the last low sea level. MIS3: indurated infralittoral marine muds, redrawn from [27].

On the outer edge, at water depths between 120 and 110 m, the first unit of the sequence (H0) is a green sand with a high concentration of glauconitic grains (60 to 80%). It is more or less ubiquitous and reaches an average thickness of 1 m, which can increase locally to 2 to 3 m. According to the remains of benthic microfauna [27], these are old littoral or circalittoral muds that have undergone significant wave winnowing. However, depending on the intermittency of the winnowing, this first accumulation presents more or less muddy or more or less sandy microsequences. This H0 is roughly contemporaneous with the last low level of 18–20,000 years BP.

The second unit of the sequence (H1) is a shelly sand, generally one meter thick, lying at water depths between 110 and 90 m. It is notably composed of bioclasts of mollusks, bryozoans, Amphisteginidae, Miliolidae, and other benthic foraminifera. The glauconitization of the shell debris is much less general than that of the fecal pellets which, to varying degrees, can be considered systematic. The abundant muddy matrix of coprolites constitutes, a priori, an ambient environment more favorable to the cationic exchanges of mineralogenesis than a poorly muddy shelly sand. A significant part of these bioclasts was a site of glauconitization, which developed in the form of infillings of chambers or other biogenic cavities, resulting in magnetic fraction contents of up to 20 –30%. This second unit, and its Amphisteginidae in particular, have been dated several times between 12,000 and 10,000 years BP (cf. previous paragraph); it logically follows the green sands of the first unit [27] (Figures 6 and 7). Several core sections allow a direct observation of this succession, but in other sites close to the middle part of the shelf, we find this second unit directly in contact with low-level alluvial deposits (MIS 2), or even Plio-Pleistocene sands, known as "Série des Cirques".

Finally, the third unit (H2) expresses the sedimentation of terrigenous mud during the last part of the Holocene transgression, particularly the high-level stand. It is a fairly rapid accumulation (4–10 $cm^{-2}$, $10^{-3}$ years), where the fecal pellets at the start of glauconitization are fairly weakly concentrated [28]. The gray or beige fecal pellets usually remained in the very early stage of glauconitization.

On the top of the slope, in a permanent marine-immersion condition, the sedimentation developed in the last hundred-thousand years is quite regular and active. There is an evolution from the highest concentrations of glauconite from the outer edge of the shelf to the top of the slope. Between water depths of −125 and −500 m, the effects of the Congo River inputs on glauconitization can still be randomly observed. More precisely, between −125 and −200 m, the alluvial sedimentation is quite rapid and masks the glauconitic deposits of the last regression up to 125 km from the river mouth. Deeper, at −300 m, the alluvial cover extends up to about 250 km from the estuary. At −500 m, there are practically no glauconitic concentrations above 1%.

We therefore retain the notion of transgressive Pleistocene and Holocene glauconitic sequences on the outer and middle parts of the West African shelves. There are short sequences, frequently condensed to only one to two meters thick, whose units could be strongly planed and reworked. The underlying compacted MIS 3 marine-mud remains provide an indication of the age of the beginning of this sequence, whose green sands and shelly sands may have been formed for at least 20,000 years and 10,000 years, respectively. As a consequence of these evolution times, the mineralogy of the 20,000-year-old green grains shows a higher rate of TeOcTe sheets and a higher $K_2O$ concentration than that of the 10,000-year-old grains.

## 6. Proposal for Glauconitization Chronometry in the Gulf of Guinea

The only Tertiary deposits of the continental shelf containing glauconite are those of the Miocene. During the long Plio-Pleistocene emersion periods, when the alluvium and colluvium of the so-called Série des Cirques accumulated, these glauconite deposits were largely altered and dispersed and could not play a role in the subsequent glauconitic successions.

Only the shoreline evolution of the Gulf of Guinea on the scale of the last glacio-eustatic cycle provides a chronological framework for the progress of glauconitization in the emerging shelf deposits, particularly on its outer edge. Chemically, glauconitization is often described as a transformation reaction of K-poor, Fe-smectite precursor to K- and $Fe^{3+}$-rich glauconite via the formation of glauconite-smectite intermediates at the sediment–seawater interface. These developments are early diagenesis processes, which concern the neoformations that develop after deposition and not necessarily after burial.

These are the successive appearances of two mineralogical assemblages: (1) ferriferous and potassic neoformed smectites, and (2) the recrystallization of these smectites to reach a more closed TeOcTe mineral. The duration of all the processes is in the order of $10^4$ years [1,2,4]. Corings on the outer edge of the Congo shelf have enabled the observation of deposits of marine mud indurated during the last emersion and containing microfauna rich in the pelagic or circalittoral components of rather warm waters (linked to the absence of the Benguela current) that are quite distinct from these of Holocene. These muds, measured by radiocarbon several times beyond 35,000 years, date back to the MIS3 episode. These are fairly strongly carbonated deposits that are almost completely devoid of glauconitic green pellets. Therefore, within the local framework of the outer edge of the shelf, we have a chronological marker that allows us to locally predate the first steps of glauconitization [27].

This mineralogical evolution is observed from the coast out to sea, where it reflects the increasing age of the sediment and, therefore, of the maturation of the grains whose stages we seek to identify [1] (Figure 8).

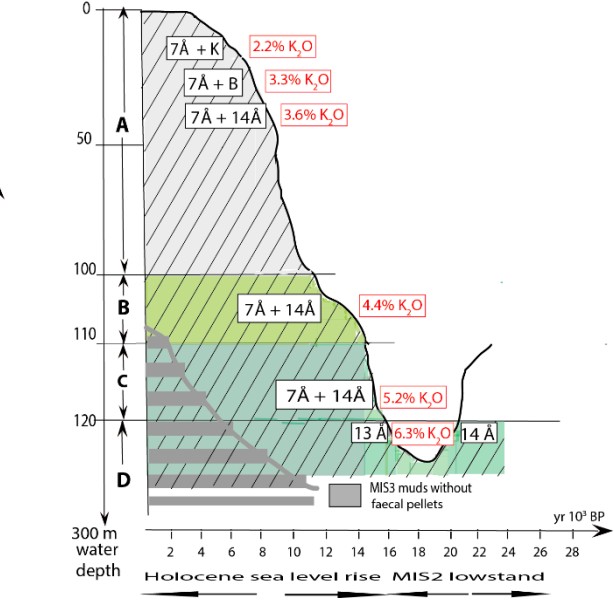

**Figure 8.** Mineralogical and geochemical evolution observed on the shelf of Congo River from the coast out to sea. An increasing age of the sediment and, therefore, of maturation of the grains is identified through phases A, B, C, and D. Partly redrawn from [1,2,27].

*Phase A.* These are the most littoral muds to have been deposited recently; they are mainly above 50 m in water depth from the active fluxes of the Congo River. A TeOc structure close to beidellite appeared over a fairly short time, estimated at4 to 5000 years. Thanks to HRTEM observations, a mean direct magnification of 100,000, shows the intimate and characteristic organization of the main phases, consisting in a spindle-shaped arrangement of layered crystallite packets (Figure 9) [29]. These crystallites are generally thin, wavy, and strongly diffracting. Some flakes show a pseudo-hexagonal tendency. The corresponding chemical analysis and apparently very weak contrast suggest this phase strongly resembles a kaolinite; elsewhere, an inhomogeneous, structurally non-organized, strongly diffracting, and Fe-rich material is visible, similar to a gel. This implies that the kaolinite-7 Å Fe phase

transformation [29] took place over time, following a dissolution–crystallization process, since there is no morphological resemblance, structural inheritance, or structural relation between them.

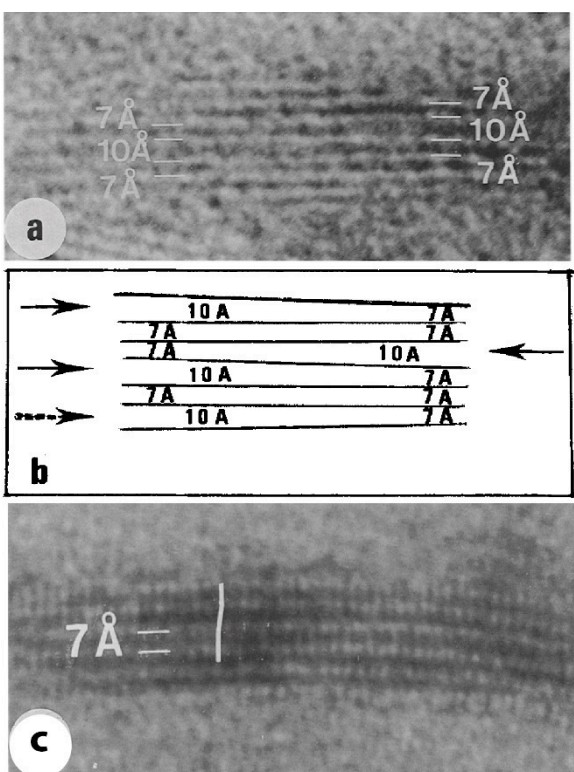

**Figure 9.** HRTEM micrographs of a richest 7 Å phase off Congo mouth. (**a**) Lattice-fringe image of a rather regular 7/10 Å interstratified structure along c. Note the lateral transformation of one layer of the 7 Å Fe phase to one layer of 10 Å-like mica phase within the (001) plane. (**b**) Schematic representation of (**a**,**c**). Structure image of one crystallite of the 7-Å Fe phase showing its faulted 1M polytypic sequence, after [29].

*Phase B.* This is the main stage of the Holocene sea-level rise, the remains of which emerge at water depths between 110 and 50 m. These are generally coastal sands of shell debris, where glauconitization is expressed by the replacement of inherited TeOc minerals by the increasingly potassium-rich TeOcTe minerals that will become dominant.

*Phase C.* The same process is more advanced in the grains of the outer edge of the plateau at water depths between 100 and 120 m, where the TeOcTe minerals become predominant and where the $K_2O$ content rises higher than 5 wt.%.

*Phase D.* At water depths below 120 m, the bottoms remained continuously submerged during the glacio-eustatic cycle. It is a low-level accumulation that formed under constant bathymetric conditions between 22,000 and 17,000 years ago. The deposit is one to three meters thick and does not show any significant vertical evolution in the mineralogy of its fecal pellets.

The final phase is that of the complete closure of the sheets at 10 Å. It cannot be observed in this framework of Late Quaternary sedimentation because a diagenetic time in the order of one million years or more is required to lead to perfect glauconites of the type 1 M [23].

These observations concern only the shelf deposits during the last glacio-eustatic cycle, since those of the previous cycles of the Pleistocene were mechanically dispersed during the low sea level.

These observations lead us to insist on the importance of the winnowing effect of the swell on the concentration and evolution of the glauconitic pellets in low sea-level periods.

In a transgressive episode, the waters are warmer and the deposits more carbonated. Moreover, a faster accumulation is unfavorable for efficient winnowing.

In the case of the outer shelf off the Congo River, the glauconitic grain is the result of successive processes. The first process is *biogenetic*: it is at the origin of concentrations that can reach up to 30 wt.% of pellets (at the scale of the whole sediment). The second is *biogeochemical* (grain scale): it allows the fixation of iron in the pellets during the oxidation of organic matter. The third is *mechanical* (sediment scale): it is levigation during regression. The last (grain scale) is geochemical: it is slow mineralogenesis in a semi-confined microenvironment. All this takes place in a regression–transgression cycle

## 7. Contourite Glauconitization

Based on several decades of research, we must add to the table of glauconitization the important processes that develop on the slopes affected by the contouritic currents.

Several investigations of contouritic deposits along continental margins revealed important glauconitic accumulations at water depths between 2000 and 3000 m [5,30–32]. Similar to continental shelf occurrences, the presence of abundant glauconitic grains in contouritic systems indicates prolonged exposure at the water–sediment interface, reflecting recurrence of sediment reworking related to the winnowing action of contouritic currents. Due to iron incorporation, glauconite grains typically display higher specific gravity (between ~2.4 and 2.9 $g/cm^3$ [33] than the average surrounding sediment (~1.7 $g/cm^3$) [34]. As a consequence, authigenic grains of glauconite are less likely to be remobilized and dispersed by currents, compared to empty foraminiferal shells.

### 7.1. Ivory Coast–Ghana Continental Margin

The Ivory Coast–Ghana continental margin is located in the eastern equatorial Atlantic Ocean (western Africa). Its southern segment bounds the Ivory Basin toward the south in a NE–SW direction and is referred to as the Ivory Coast–Ghana Marginal Ridge. It is 130 km long and 25 km wide. This very steep transform margin experiences documented interference from bottom-water circulation.

The ODP Site 959C depth corresponds roughly to the interface between North Atlantic Deep Water and South Atlantic at a water depth of 2100 m, at a moderate distance from the shore (~120 km) and a seawater temperature near 3 °C [5,13,35]. Its sediments are of particular interest because of their generally slow (~12 cm/kyr) accumulation rate from at least the Miocene to the Holocene [35]. The uppermost 25 m consists of thin dark-gray-to-greenish sedimentary layers. The green clay material usually fills the chambers of the globigerine or buline or some other pelagic foraminifers. Other grains correspond to the internal molds of the radioles of holothurids. Ellipsoidal fecal pellets only are rare or absent. This material is completely devoid of any littoral benthonic evidence, as Miliolidae or Amphisteginidea, and cannot be interpreted as perigenic, i.e., passing from the outer shelf to the bottom of the slope.

The green-grain concentration does not appear to be constant over time, suggesting low content (<10 wt.%) below 12 mbsf and higher contents above this depth. In particular, the interval ranging from 12 to 11 mbsf presents the highest green-grain concentrations (20%–45 wt.%). Above 11 mbsf, the green-grain contents are commonly higher than 10 wt.%. The maxima concentrations seem to be associated with higher values of the green–white ratio (medium–dark green/white–pale green) and, to a lesser extent, to the relative abundance of the cracked dark-green grains.

Generally, the lower contents of green grains correspond to white-to-pale-green infillings that are characterized by lower potassium content (1.4%–3.1 wt.% $K_2O$), an irregular content of iron (29.7%–42.7 wt.% $Fe_2O_3$), and a slightly high alumina content (7.2%–8.9 wt.% $Al_2O_3$). According to a previous Mossbauer analysis in the Gulf of Guinea, the $Fe^{2+}/Fe^5$ ratios are between 0.08 and 0.24, a ratio that increases in line with the growing proportion of associated goethite. The highest $Fe^{3+}$ contents correspond to the shallowest sediments, where goethite and quartz grains are more abundant [25]. By contrast, higher potassium

and, roughly, iron contents are related to green-to-dark-green infillings, which are sometimes more or less split by fissures and cracks (4.6%–6.8 wt.% $K_2O$, 45.6%–52.7 wt.% $Fe_2O_3$, and 3.4%–5.2 wt.% $Al_2O_3$, respectively). In the same levels, green-to-dark-green grains with deep cracks are widespread and sometimes abundant, especially in the 11–12 mbsf interval. The contents obtained were as follows: 7.1%–8.8 wt.% $K_2O$, 44.7 wt.%–55% $Fe_2O_3$, and 2.5–3.3 wt.% $Al_2O_3$ [5].

Based on petrographic, mineralogical, and geochemical analyses on separated bulk green-grain fractions [13], the authigenic nature of the green clay minerals in these sediments was established and their overall composition was determined to be dioctahedral $Fe^{3+}$-montmorillonite, with minor proportions of interstratified Gl-Sm (~20% glauconite layers and ~80% Fe-smectite layers) in the highly evolved dark-green grains.

Each contourite sequence is characterized by a high abundance of glauconitic grains (especially in the moat area), but also in moderately to strongly discontinuous sediment accumulation associated with relatively low sedimentation rates. Usually, this abundance contrasts with the scarcity of nearby deep-sea settings that have experienced continuous fine-grained hemipelagic sedimentation. This contrast is observed during the Late Quaternary along the Ivory Coast–Ghana Ridge and the French Guyana margin [13,15].

### 7.2. Demerara Margin (French Guyana)

Along the Demerara margin, at a water depth of 2400 m, the occurrence of both higher concentrations of glauconitic grains and increasing relative abundances of evolved dark-green grains during MIS 2 has been interpreted as reflecting a strengthening of bottom-water circulation over glacial time [15]. Lippold et al. [36] evidenced the persistence of a vigorous oceanic circulation during the LGM in the relatively shallow South Atlantic, but the case of the Demerara Plateau is even more peculiar because of its particular morphology: the large indentation is thought to be at the origin of the intensification of the NADW, favoring the formation of strong contouritic currents.

Since the studied site is located in the core interval of the modern NADW, this finding was taken as evidence of an intensification of the glacial AMOC [8,15]. These elements suggest that the glauconitization might be related to higher winnowing intervals. Moreover, the presence of the glauconitic facies described here corresponds to sediments collected in the stronger bottom-water winnowing, whereas this facies is absent from the sediments collected in the drift (weaker bottom-water winnowing, finer sediments, and rare glauconite fracturing features).

As previously suggested [4,5], the SEM-EDS data indicate that the glauconitic grains progressively evolve towards darker green shades as they incorporate higher amounts of Fe and K. Successive glauconite neoformation steps are also accompanied by a gradual decrease in Al contents, generally reflecting the gradual disappearance of inherited terrigenous TeOc minerals, such as kaolinite. By contrast, the corresponding increase in $K_2O$ and $Fe_2O_3$ (note that the $Fe^{2+}$ content is generally less tan 20%) is largely independent of the presence of inherited minerals. The K in glauconite is directly derived from ambient bottom water, sequestered between newly formed and/or transformed micaceous sheets. The abundance of $K_2O$ thus reflects the process of glauconitogenesis [13]. The increase in $K_2O$ contents, and, by extension, $K_2O/Al_2O_3$ ratios, most likely corresponds to the presence of neoformed sheets of TeOcTe clay minerals, such as K-bearing montmorillonite and illite.

These roughly summarized processes therefore indicate glauconitization that is broadly similar to that described on the outer edges of the Gulf of Guinea shelves (see above). The complete glauconitization process (with more than 8 wt.% of $K_2O$) needs a longer period of time, ranging from 2 to 10 million years [4,6,18], and, therefore, often occurs during late marine diagenetic processes.

## 8. Schematization of Cation Exchanges during Pellet Glauconitization

Tropical marine muds are favorable to glauconitization because of their richness in iron (from 10 to 20 wt.%) and organic carbon (more than 3 wt.%). In a first step, the strongly anoxic sedimentary environment of organic mud accounts for the mobility and solubility of terrigenous iron from the $Fe^{2+}$ state in which it is transported. In a second step, the pellet is an oxidative micro-environment, in which $Fe^{2+}$ is rapidly oxidized to the $Fe^{3+}$ state and trapped. The first enrichment of the pellet is accompanied by aluminum and silicium depletion in inherited minerals, together with a significant enrichment of iron and potassium in neoformed minerals [3]. The K in neoformed clays is supplied by ambient seawater.

In Phase 1, the fecal pellet is subject to dehydration, allowing early porosity (through pores and inner and outer cracks) and, therefore, early oxidizing microsites. Recent work has shown that in small pores, part of the water is attached to the wall, and the water's activity is diminished; the precipitation of a great number of mineral species is thereby made easier, and their stability domains are changed [4]. The organic matter is rapidly mineralized; the C org content is thus divided by 2 or 3. This early oxidation of organic matter causes a redox front between the granular support and the surrounding mud, making the micro-environment inside the foraminifera more oxidizing than the mud matrix [4,5,10,13]. Therefore, from the start of diagenesis, the pellet functions as a trap for $Fe^{3+}$, which then concentrates very quickly and induces the precipitation of more or less well crystallized iron oxy-hydroxide aggregates, which do not intervene in glauconitization and, therefore, cannot be taken into account in the geochemical analysis of the subsequent processes. Another part of iron ($Fe^{2+}$ or $Fe^{3+}$) participates in the transformation or neoformation, which intervene in one another during glauconitization, i.e., in the center of the octahedra of Te-Oc structures quite close to beidellite or later Te-Oc-Te structures of the $Fe^{3+}$-montmorillonite type. Generally, glauconitization implies a partial reduction in iron in the sedimentary materials of pellets [37–39]; although a noticeable portion of the iron in glauconites is in the divalent state, there is no clear relationship between the abundance of the two valences in glauconites.

This early oxidation of organic matter inside the pellet causes a redox front between pellet's micromilieu and the surrounding mud, making the inner micro-environment of the pellet more oxidizing than the mud matrix [3,4,13,30]. Thus, $Fe^{2+}$ migrates rapidly inside foraminifera tests, where it is partly and quickly oxidized into $Fe^{3+}$ or, possibly, incorporated into octahedral structures; smectites become Fe-smectites [13,14,18].

In these nascent and slightly evolved stages, it is the inherited minerals that are still dominant in the composition, i.e., kaolinites in various process of degradation, illites or a few muscovite flakes and small quartz less than 50 µm in diameter (Figure 10). All these minerals disappear fairly quickly during glauconitization, with the quartz remaining the longest. In contrast to Fe, K migrates slowly and gradually from seawater inside the micro-environment to form glauconitic assemblages. The K begins to penetrate into the microenvironment, where it is destined to occupy the interfoliar spaces of the first newly formed sheets. Incidentally, we also note the parallel movement of magnesium from seawater to semi-confined pellets [13,18,25].

In Phase 2, we observe the appearance of large cracks on the surfaces of the pellets, the greening of which intensifies (transition from medium green to dark green). Several mineralogical steps follow one another with growing proportions of $Fe^{3+}$-montmorillonite at 13 Å, or of nontronites [18]. In the most advanced stages, sheets closed at 10 Å indicate the transition to glauconites. All of these steps occur with fairly constant or only slightly increasing iron contents.

Phase 1

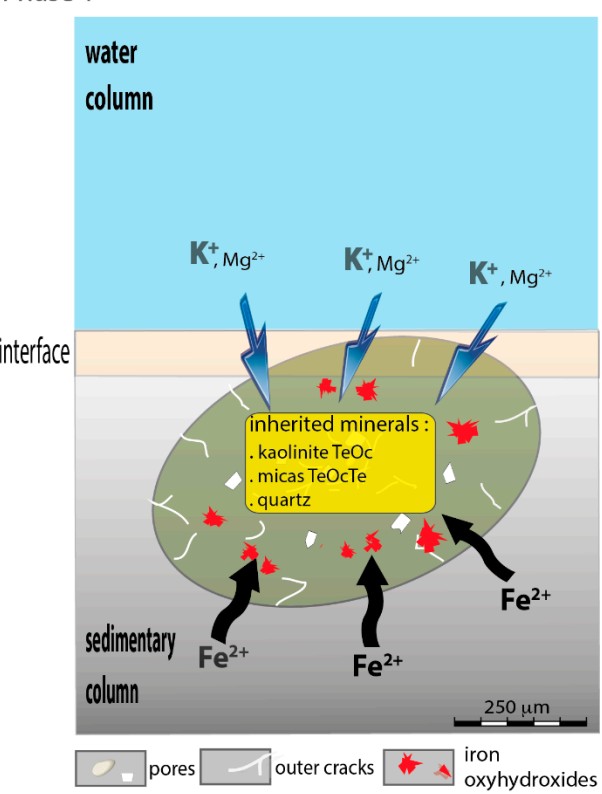

**Figure 10.** First phase of schematic evolution of glauconitic grains from pristine marine mud to pale–medium-green grains with the mean elementary composition measured by microprobe and established for each class color.

The interlayer incorporation of K and, consequently, glauconitization, requires a long exposure period at the seawater–sediment interface. If these two conditions are met, the transformation of detrital clay minerals into $Fe^{3+}$-smectite, interlayered $Fe^{3+}$-smectite/glauconite, and glauconite is possible (Figure 11). The formation of mature glauconitic grains, composed of $Fe^{3+}$-smectites and interlayered $Fe^{3+}$-smectite/glauconite (with 5%–6.5 wt.% $K_2O$ in our samples) occurs rapidly during the early diagenetic process (about 10–100 thousand years).

On the other hand, $K_2O$ and, more incidentally, MgO, continue to concentrate in the microenvironment, where they participate in the neoformation of TeOcTe minerals. The thermodynamic conditions lead to the complete disappearance of alumino-silicates and inherited silicates; the departures of aluminum and silica express this degradation. As stated above, in contrast to Fe, K migrates slowly and gradually from seawater into the micro-environment (Figure 11). The interlayer incorporation of K, and, consequently the glauconitization, requires a long exposure period at the sea water–sediment interface. If this condition is met, the transformation of detrital clay minerals into $Fe^{3+}$-smectite, interlayered Fe-smectite/glauconite, and glauconite is possible [5]. The pellets become more cavernous and promote the "nesting" of clusters of newly formed sheets. At the same time, the replacement of the TeOc structures by highly hydrated TeOcTe structures lowers the density of the whole pellet, the surface of which cracks significantly, allowing the mechanical fragmentation of these particles during subsequent reworkings. During the end of the Pleistocene ($10^5$ years), the microenvironment approached relative thermodynamic stability; the total completion of the process required a long process of diagenetic evolution, which can only be found in the Cenozoic deposits. Burial separates synsedimentary geochemical reactions (the type with which this paper is mostly concerned) from diagenetic reactions.

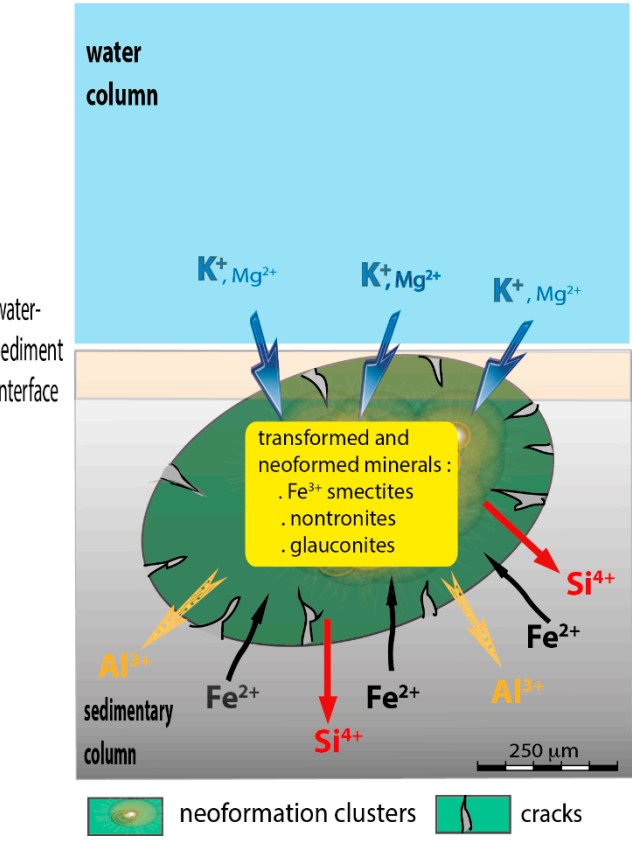

**Figure 11.** Second phase of schematic evolution of glauconitic grains from pale–medium-green grains to dark-green grains with the mean elementary composition measured by microprobe and established for each class color with the two values on each grain.

## 9. Discussion and Perspectives

### 9.1. Comparison of Glauconitic Sequences on Shelf Versus Contouritic Slopes

Due, mainly, to iron incorporation, glauconite grains typically display higher specific gravity (between 2.4 and 2.9 g/cm$^3$ [33] than the average surrounding sediment (1.7 g/cm$^3$) [34] As a consequence, authigenic grains of glauconite are less likely to be remobilized and dispersed by currents than empty foraminiferal chambers.

#### 9.1.1. On the Gulf-of-Guinea Shelf

Core sections make it possible to observe the vertical succession described from the various outcrops on the shelf. On the outer edge, various exposures of the last regression can be observed at the outcrop. The sections show alternations of glauconitic mud (5 to 10 wt.% green grains) and muddy glauconitites (50 to 80 wt.% green grains).

These deposits were formed at a short distance from the 110-meter-water-depth shoreline (Figure 12a):

- Glauconitic concentrations correspond, as we have seen, to levigation phenomena by the swell; these are glauconitic coastal strips;
- Muds accumulate in sheltered depressions where the smectite contents of the green grains are higher (six to eight parts in ten) in the sandy deposits, but they are lower (four to six parts in ten) in muddy deposits, i.e., glauconitization is less advanced where burial is faster (Figure 12b). Where the coastal bars are the thickest, a positive correlation is still observed between the contents of green grains (up to 20%) and smectites (up to six parts in ten).

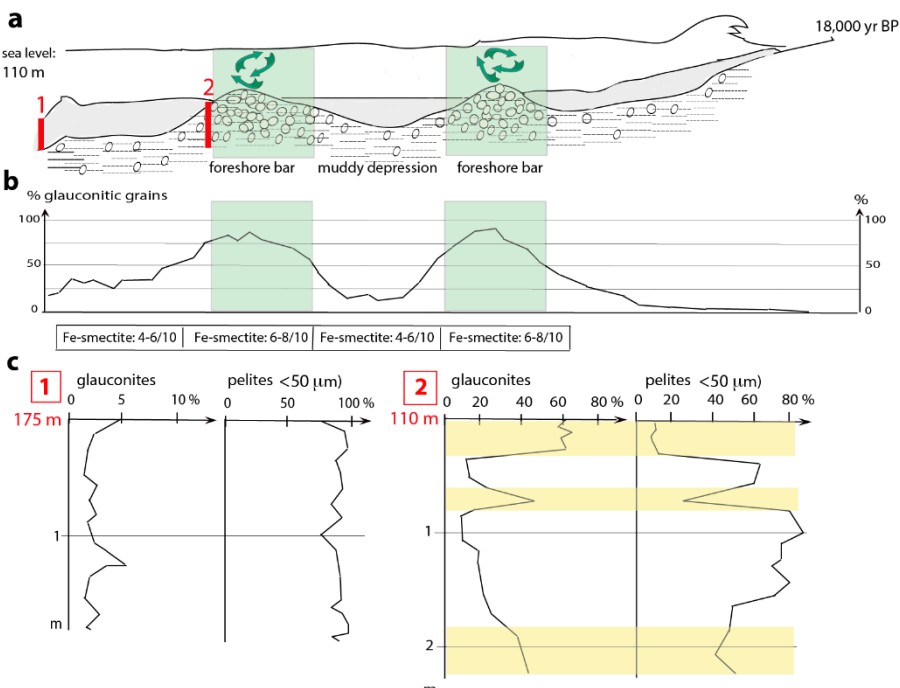

**Figure 12.** (**a**,**b**) Simplified diagram of the deposition and levigation processes of glauconitic sand barriers around 18,000 years BP on the Gulf-of-Guinea shelf, with locations of sediment cores 1 and 2 at depths of 175 and 110 m, respectively; (**c**) 1. at −175 m, from a distance of the 18,000-years-BP shoreline, the glauconitic accumulation is low and regular; the glauconitization process is more advanced towards the top of the deposit (end of the Holocene). 2. Alternation of deposits with high (pale-yellow bands) and low energies at −110 m during the 18,000-years-BP regression; green grains are the best crystallized and concentrated in the sediment. Alternation of deposits with high and low energies at −110 m during the 18,000-years-BP regression; the green grains are the best-crystallized as they are numerous in the sediment (partly after [27,28].

On the top of the slope, where the muddy sedimentation is slightly homogeneous over the entire vertical (Figure 12c), the green-grain concentrations remain low and almost constant, while their mineralogical composition hardly changes.

In conclusion, a more thoroughly achieved mineralogical evolution controlled by burial cannot be demonstrated on this shelf. On the other hand, within deposits of the same age, differences in evolution can be observed in connection with changes in facies attributed to different rates of sedimentation/erosion. The duration of grain exposure to the water–sediment interface controls the progress of mineralogical evolution.

### 9.1.2. On the Slope

On the slope of the Ivory Coast–Ghana Marginal Ridge, the Pleistocene pronounced low sedimentation rates (1 to 2 cm/kyr) most likely reflect the persistent activity of deep-water circulation. It is therefore suggested that these low rates were largely controlled by intensive winnowing and sediment redistribution, caused during an enforced bottom-current episode.

Based on the oxygen isotopic record of the 959 ODP Site (2100 m water depth), two major stratigraphic gaps are reported at ~11–12 mbsf (ca. 520,000 years BP) and at ~1.5 mbsf (ca. 85,000 years BP), as a consequence of enhanced bottom-water circulation [5]. These two scoured (eroded) surfaces may have resulted from the same paroxysmal dynamic process; the deeper gap was observed just above the major hiatus of the stratigraphic column, at the end of an erosional discontinuity. All these green-grain concentrations (10%–40 wt.% of the sand fraction, 0.5%–15 wt.% of the bulk sediment) are unusual deeper than 2000 m. In this case, the winnowing associated with low detrital input produced

decreased accumulation rates that prevented an excessively rapid burial. The grains were exposed at the sediment–water interface for periods sufficient to allow glauconitization.

A detailed microstructural analysis indicates a number of laminae 1 mm to 1 cm thick with increased concentrations of foraminifers. Many more glauconitic infillings of foraminifers were observed ca. 1–0.9 Ma, just after the major hiatus (Figure 13a). Assuming that the superficial redistribution of the sediment was favorable to longer contact with seawater and to mineralization, the green-grain concentration supports the indication of the long duration of an uneven beginning of sedimentation after the hiatus.

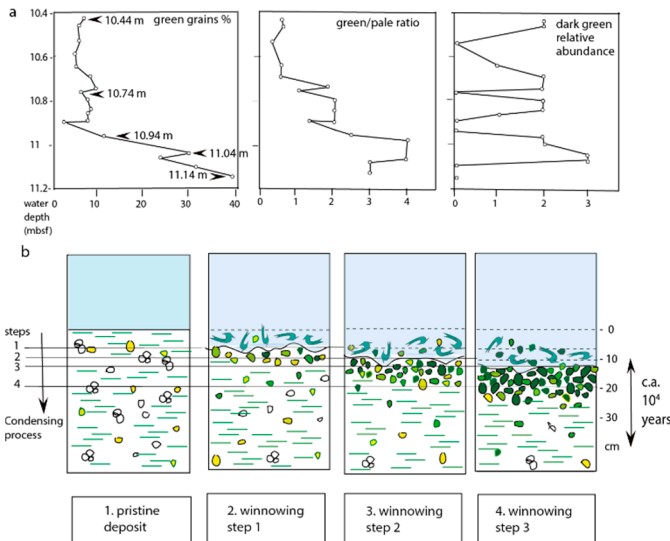

**Figure 13.** (**a**) 959 ODP Site. Example of green-grain vertical distribution between 11.2 and 10.4 mbsf: green-grain concentration, green–white grain ratio, dark-green (cracked) grain relative abundance. (**b**) Step-by-step condensation process of the green grains. Bottom-current-controlled green-grain deposition and evolution, based on mud-matrix reworking, green grains' mechanical concentration, and mineralogical maturation, reflecting the time of residence of the grains at the sea bottom before burial. Foraminifer tests were gradually removed or destroyed from 1 to 3, after [5].

Many more glauconitic infillings of foraminifers were observed at ~11–12 mbsf, probably accounting, at least in part, for the interval's low accumulation rate. The usual nannofossil ooze with foraminifers is succeeded uphole by a winnowing 1-centimeter-thick lag of foraminifers with some light-green infillings.

An example of the microsequence of the 959 ODP Site between 11.2 and 10.4 mbsf in depth shows the correlation between the concentrations of green grains and the ratio of evolved green grains to pale or white grains at the start of glauconitization. The interval of the microsequence between 11 and 10.4 mbsf corresponds to quieter waters favorable to sedimentation [5]. Such stable waters do not prevent forms of reworking of dark-green, highly-evolved grains, which can still be observed towards the top of the microsequence (Figure 13b). The concentration of dark-green grains at the base of the microsequence occurs under the control of a paroxysm of winnowing, which gradually results in a condensed microsequence. The sandy particles concentrate to the detriment of the fine or light particles, which are resuspended and transported away. These particles are mainly foraminifera shells whose chambers are filled with mud, which support glauconitization. Their fate improves as a result. The final phase of the winnowing action is an accumulation of dark-green grains with high density (>3).

Even at the deeper site at the Ridge, Site 962 (4637 m water depth), sediment accumulation was also interrupted by numerous hiatuses, and the presence of several glauconitic hardgrounds suggests that the winnowing process of Site 959C was probably enhanced.

Thus, as previously indicated, the winnowing process is the cause of the long-lasting ionic exchange and mineralogical evolution at the water–sediment interface. In these

marine tropical deposits, the glauconitization process was especially favored by abundant iron supply and was not affected by low water temperatures (around 3 °C).

It is suggested that the green-grain concentration was not exactly a synsedimentary processes, but was controlled by the intensity and duration of secondary winnowing and by the duration of diagenetic cationic exchanges with overlying fluids (controlling the progress of mineralogic evolution of the grains at the sediment–water interface). In this assumption, the fluctuations in green-grain concentrations correspond to microsequences linked to the pulses of energy of bottom-water. The thicknesses of individual green grains expressed at this section between each concentration peak are slightly variable, ranging from 15 to 50 cm, with an average thickness of 38 cm.

In Despite its deep-water position, the green-grain concentration can be estimated as adequate for the glauconitization process. Pyrite was commonly observed as individual framboids or as late overgrowths on the glauconitic infillings. This mineral expresses iron trapping; its presence is apparently opposed to that of glauconitic grains.

*9.2. Temperature Incidence on Recent Glauconitization*

A common factor in the recent glauconitization that was previously and widely invoked is temperature. The synsedimentary marine green clays formed are glauconitic minerals; they develop in cold (less than 10–15 °C), slightly deep waters (over 60 m water depth) and sheltered from continental detrital outflow to the sea [4]. On the Nigerian shelf, Porrenga [40] considered that the temperature factor is sufficient to distinguish a favorable environment for 7 Å neoformation (temperature 25 °C) that is especially characteristic of the glauconitization process (temperatures above 11 ± 4 °C). The same range of temperatures was reported from various latitudes: 7 °C at high latitudes off Norway, or 14.3 °C at low latitudes a depth of −300 m in the Makassa Strait [4].

These temperatures, which mostly concern shelf or upper slope sites, are generally higher than the 3 °C reported for the 959 ODP site on contouritic and glauconitic slopes (2100 m water depth) of the Ivory Coast–Ghana Marginal Ridge [36]. However, on the same slope, this water depth does not appear as a discriminating factor because, in close to the 962 ODP site, at a 4637-meter water depth (probably at 2 °C), sediment accumulation was punctuated by "glauconitic" concentrations, even green hardgrounds, suggesting that bottom currents were active at least at this water depth. More recent work on the Demerara margin, which was exposed for several million years to contouritic currents, completes our knowledge of deep glauconitization, observed at water depths between 2370 and 3214 m [8].

Thus, access to the deep oceanic domain, that of contourites, makes the concept of fairly-high-temperature waters, which has prevailed for a long time, obsolete. At the same time, it somewhat relativizes the notion of the direct influence of the nature of continental flows and, in particular, that of iron. Glauconitization can develop in deep tropical waters, even at long distances from the mainland.

On the other hand, the restitutions of the paleotemperatures of the Mesozoic or Cenozoic deposits where green grains underwent a long diagenesis after burial lead to very different conclusions. From the oxygen isotopic signatures of a calcareous substrate, a formation temperature of 26 ± 2 °C has been reported for the Langenstein Cretaceous sequence [41], which is typical of the warm shelf settings of the Cretaceous. Such rapidly forming glauconites can indeed represent a reliable and robust mineral archive, which is applicable to geochronological age dating, paleo-reconstructions, and regional correlation of localities.

*9.3. Iron-Determinant Intervention and Pleistocene Timing of Glauconitization*

These observations raise the question of the specific function of iron in the intertropical processes of glauconitization on the two outer shelves and contouritic slope environments of the Gulf of Guinea and Guyana margins. In both cases, iron is terrigenous, but according to the increasing distance from the source, the accumulations decrease consistently seaward.

However, and despite the slightly moderate iron concentrations of the sediment, glauconitization is very active in the contouritic deposits of both the western (Ivory Coast–Ghana) and eastern (Demerara) margins of the tropical Atlantic. In both cases, highly evolved green grains lead to mineralogical compositions strongly dominated by glauconite sheets with high iron and potassium contents.

Out of intertropical latitudes, iron fluxes are restricted by lower hydrolysis on the continent [42], and glauconitization processes are less active [43]. There are few studies of glauconitization at these medium latitudes because the low levels of glauconite accumulation have limited the analytical possibilities during recent decades. Despite fairly low concentrations, however, glauconitization shows stages of evolution that can be quite advanced, as in the Gulf of Lions [10] and Kerguelen [44], and, apparently, comparable with those of intertropical latitudes. However, the main difference is in the total concentration of glauconitic grains in the sediment, which is significantly lower than those in intertropical oceanic environments. Consequently, these examples suggest that a relative iron deficiency restricts the growth of newly formed glauconitic masses.

Glauconitization is a comparatively long process. If it is accepted that a few thousand years are needed for the formation of high amounts of Fe-smectite, which may need some $10^5$ or $10^6$ years to develop entirely. Within the scope of these limited case studies related to the end of the Pleistocene period, we were able to record many of the stages of glauconitization, but not all of them, since the most closed glauconitic sheets generally indicate $K_2O$ contents lower than 6 wt.%. On the shelves of the Gulf of Guinea, glacio-eustatic movements make it possible to frame the duration of glauconitization: most of its evolution was carried out on the scale of a single regression–transgression cycle, i.e., approximately $2.10^3$ years. Since the glauconitic sequences of the contouritic slopes of Ivory Coast–Ghana and Demerara were not affected by glacio-eustatic movements, the dating is difficult by conventional methods due to the frequent erosion of the top of the deposits and recurring reworkings; however, the example of the glauconitic sequence around 11 mbsf from the ODP 959 Site of the Ivory Coast–Ghana Ridge allows us to suggest that a time interval in the order of $10^4$ years is necessary for its accumulation and mineralogical evolution.

A few thousand years are needed to form sufficient amounts of Fe-smectite, which subsequently matures into glauconite at different timescales, which may range from between ~5 and >10 Ma for cold, deep-water settings to <1 Ma for warm, shallow-water areas [45].

### 9.4. Comparison of Glauconitization on the Shelf and on the Deep Slope

In a broad outline and according to more recent studies, glauconitization near the water–sediment interface is controlled chiefly by the sediment substrate, such as fecal pellets or foraminifera chambers, which provides semi-confined micro-environments both in calcareous and siliciclastic sequences [46–48].

On the basis of this overview, I consider that the factors determining deep glauconitization are similar to those that affect the slightly higher outer shelves, enhancing the verification of the coherence of the processes. However, transport and contouritic deposition contribute to specificities of deep-water glauconitization. This is linked to the paroxysmal energy of the contouritic currents' winnowing action, which is at the origin of the glauconites that are formed and accumulate at water depths of more than 2000 m. On certain deep slopes, agitation by contouritic currents, by increasing the duration of the interaction at the water–sediment interface, repeatedly provides the conditions necessary for multisequential glauconitization.

It is suggested that in deep-water environments, bottom-current-controlled iron deposition and concentration enhances mineralogic evolution. Locally, the excessively lengthy dispersal of iron microparticles plays a decisive role in montmorillonite, rather than nontronite development [13]. Such cases are most commonly found in shallow-water environments.

In the first simple microscopic approach, the populations of glauconitic grains in the deep domains and the coastal domains presented several similarities. In both cases, we noted the presence of a succession of pigmentations, ranging from gray or light green to dark green, which correspond to the stages of the progression of glauconitization according to Odin's classification: nascent, slightly evolved, evolved, and highly evolved [4].

However, quite significant differences must be noted:

(1) At the level of the shape that is still identifiable in the initial stages, the glauconitic grains of the shelf appear in very diverse facies: the fecal pellets of limivorous organisms, the internal molds of benthic or pelagic foraminifera, the epigenies of sea-urchin radioles or bryozoans, stacks of micaceous sheets, and the reworked clay clasts of muddy bottoms (intraclasts or extraclasts). By contrast, the deep glauconitic grains (e.g., contouritic deposits) come mainly from the fillings of foraminifera chambers (here mostly pelagic). These infillings quickly detach from each other, becoming difficult to recognize. On the slopes, fecal pellets are only rarely observed; the unstable environment of the perpetually reworked contouritic accumulation does not favor the development of a limivorous infauna.

(2) Depending on this initial support diversity, significant differences can result in the average sizes of the glauconitic grains. The grains of the outer shelf show quite varied dimensions, which are predominantly those of benthic shell debris and fecal pellets, ranging mainly between 300 and 500 μm, and sometimes up to 1000 μm. The grains of the contouritic deposits are markedly related to the infillings of the chambers of the pelagic foraminifera, which are isolated by the repetition of the mechanical actions; their average diameters range from 150 to 300 μm.

(3) Concerning color, the glauconitic grains of the continental shelf present the complete range of stages of progression of greening, but also beige, ocher or brown grains, which attest to the erratic presence of ferric iron in the composition and which, eventually, demonstrate the associated of oxidation processes with episodes of emersion. This oxidation can lead to high concentrations of iron inside the grain without participating entirely in the glauconitic neoformation; this iron is locally concentrated in the amorphous state, or even crystallized in the form of goethite aggregates [13]. In these cases, the concentration of iron determined in the bulk analyses could be taken into account unduly in the calculations of the structural formulas and become a source of analytical bias. This drawback cannot exist in the grains of deep contouritic deposits, where the risk of oxidation is absent. Indeed, ocher grains are never observed in the glauconite populations of contouritic deposits.

In deep-water settings, glauconite formation is believed to be extremely slow, requiring up to ~9 Ma to form >90% glauconite layers, which is mainly due to the low temperature (<5 °C) of the deep ocean waters and the limited or discontinuous supply of the rate-limiting elements, such as Fe. The glauconite grains forming in such cold water are often characterized by a heterogenous composition [46–48]. It was suggested that deep waters are linked to slower sedimentation rates and the related decreased influx of reactive chemical components needed for glauconitization or interrupted elemental diffusion paths within the micro-environment, thus leaving immature glauconite or glauconite-smectite in the sedimentary rock record. By contrast, mature glauconite grains with a homogenous chemical composition and comparatively small age variation may form in the shallow-water settings of present-day and ancient oceans due to the enhanced influx of more reactive terrigenous components [49]. However, these assessments are based on buried Mesozoic or Cenozoic glauconite deposits, in which it is difficult to separate the respective processes developed near the water–sediment interface (early diagenesis) from those controlled by the temperature rise after burial (late diagenesis). Our observations, which were limited to the glauconitization of the last tens of thousands of years, lead to the conclusion that the conditions of tropical ocean waters are equally favorable regardless of their distance from the coast or their depth, i.e., their temperature. The last steps from Fe-smectite to glauconite end-members belong to late diagenesis and are likely to vary according to local geothermal

conditions, although they are outside the scope of this study. Another explanation could be related to the seawater chemistry of some ancient seas, such as those of the Late Cretaceous, where shallow-water glauconitization was much faster than modern rates of glauconite formation in shallow vs. deep-sea environments [45].

*9.5. Neodymium Isotopes in Glauconite as a Tool for Palaeoceanographic Reconstruction*

Glauconite grains could record ambient bottom-water neodymium (Nd) isotopic compositions and, hence, be used as paleoceanographic archives. Nd isotopic compositions were analyzed in a series of glauconitic grains, foraminiferal assemblages, leached Fe-Mn oxyhydroxide phases, and detrital clays separated from a contourite sediment record at the Demerara slope off French Guyana (IG-KSF-11; 2370 m water depth) [7]. Nd isotopic measurements were previously acquired on uncleaned foraminifera from a nearby sediment core collected at a water depth of 947 m, bathed in the present day by Antarctic Intermediate Water (AAIW) [50].

The upper 10 cm of core IG-KSF-11 is assumed to correspond to the Holocene period, thereby covering a period of time during which bottom-water $\varepsilon$Nd signatures are expected to have remained near to present-day values. In marked contrast, both uncleaned foraminifera (10.9–0.1; 0–1 cm core depth) and leached sedimentary Fe-Mn oxyhydroxide fractions (9.2–0.2; 4–5 cm core depth) from the same upper-core sediment depart significantly from the expected North Atlantic Deep Water (NADW)-like Holocene seawater signature. Neodymium isotopic measurements on uncleaned foraminifera are usually assumed to reflect the signatures of associated Fe-Mn oxyhydroxide coatings that precipitate onto and within the foraminifera tests at the seawater–sediment interface [51,52], hence acquiring the $\varepsilon$Nd composition of ambient bottom waters.

These results suggest the strengthened bottom-water circulation of the glacial analogue of NADW at this particular location and water depth, with a Nd signature (between $-10.8$ and $-11.5$) similar to that of the modern NADW. The $\varepsilon$Nd Holocene and the LGM ($\varepsilon$Nd~$-10$), while revealing pronounced $\varepsilon$Nd excursions towards unradiogenic values (between $-11$ and $-12$) during the short-lived North Atlantic cold periods of the Heinrich Stadial 1 and the Younger Dryas, were interpreted as reflecting a strong reduction in the AMOC, leading to the reduced influence of northward-flowing AAIW in the equatorial Atlantic [52].

The Nd composition of core-top glauconite grains ($-12.0 \pm 0.5$) agrees with the expected NADW-like seawater signature at the same location and water depth ($-11.6 \pm 0.3$), while departing from the Nd values measured for corresponding detrital clays ($-11.3 \pm 0.2$), foraminiferal ($-10.9 \pm 0.2$), and Fe-Mn oxyhydroxide fractions ($-9.2 \pm 0.2$). This finding indicates that the glauconitic grains at this particular location are probably better-suited to paleoceanographic reconstructions than foraminifera shells and leached Fe-oxyhydroxide fractions, which appear to be influenced by sediment redistribution and the presence of terrestrial continental Fe-oxides, respectively.

Considering the Nd distribution along the water column at the Demerara Rise [50], I infer that the foraminifera assemblages encountered in the contourite moat at site IG-KSF-11 may be derived from shallower depositional environments, at water depths between 600 m and 800 m, bathed by AAIW.

Overall, these new results suggest that the application of Nd isotopes to glauconite grains could serve as useful proxies for paleoceanographic reconstructions at continental margins, wherever intense winnowing and/or erosional processes prevents the use of other, more conventional archives of past seawater Nd compositions, such as uncleaned foraminifera. In particular, the combined use of Nd isotopes (as tracers of water mass) and various elemental ratios, such as $Fe_2O_3/Al_2O_3$ and $K_2O/Al_2O_3$ (as tracers of the degree of glauconitization), could provide complementary information on both the sources and the strength of past bottom-water circulation.

## 10. Conclusions

Below continents whose soils are subject to significant hydrolysis, the intertropical seabed is fed by large rivers, vectors of flows rich in iron and organic matter that determine the quality and quantity of glauconitization. From this point of view, the bottoms of the continental shelves and the contouritic slopes of the Gulf of Guinea offer favorable experimental conditions for approaching the parameters that control the processes. These are real glauconite factories, whose production and chronology can be observed at scales of $10^3$–$10^4$ years, and which are not accessible to geologists, particularly to stratigraphers, who analyze the glauconitic phenomenon in the Meso-Cenozoic period. On the western margin of the Atlantic, the interferences of the considerable liquid flow of the Amazon with the ocean currents do not prevent glauconitization, but shift or dilute its effects. However, the French Guyana margin offers a South American analogue of the glauconitic sequences in the Gulf of Guinea.

The shelves of the Gulf of Guinea present exposures of glauconitic deposits interstratified with vestiges of low-level continental episodes. These environmental occurrences allow the definition of more or less complete glauconitic sequences of the latter that can, with the help of radiocarbon, be dated directly. Glauconitization develops on a time scale of 20,000 years.

The contouritic deposits of the slopes of the Côte d'Ivoire–Ghana Ridge and the Demerara margin, despite some scoured surfaces and various consequent hiatuses linked to the paroxysms of the contour currents, can also be dated thanks to radiocarbon, but also thanks to the oxygen isotope record or paleomagnetism. Here, the glauconitic process can be perpetuated recurrently over several hundreds of thousands of years, but the glauconitic sequence event generally lasts 30,000–60,000 years.

However, the factors determining deep glauconitization are similar to those that affect the slightly higher outer shelves, enhancing the verification of the coherence of the processes.

A more thoroughly achieved mineralogical evolution controlled by burial cannot be demonstrated on this shelf.

The Gulf of Guinea constitutes one of the most favorable oceanic zones for the geochemical study of the semi-confined microenvironments of pellets that become glauconitic grains. The rapid concentration of iron within these pellets is controlled by the oxidation-reduction front at the interface of this grain with the surrounding mud. This notion of micro-confinement, already introduced in studies at the end of the 20th century, has been clarified and extended with the application of new geochemical methods (punctual analyses with microprobes and HRTEM imagery), but also through diffractometry, with the passage from the analysis of populations of grains to that of a single grain or a micropart thereof.

Thanks to the discoveries of deep glauconitic sequences in waters of 2 to 3 °C, the high temperature of tropical waters, long considered an essential factor in glauconitization, is no longer a prerequisite.

The richness in iron of the matrix muds of glauconite is recognized as essential to the processes of neoformation at various latitudes. The iron fluxes are sufficient even at a considerable distance from the continent, as shown by the glauconitic sequences of the oceanic slopes. In fact, these flows are essential to the accumulation of large masses of green grains because the stages of glauconitization can be expressed in the marine muds of temperate latitudes, where the iron is, however, less concentrated.

The combined use of Nd isotopes (as tracers of water mass) and various ratios, such as $Fe_2O_3/Al_2O_3$ and $K_2O/Al_2O_3$ (as tracers of the degree of glauconitization), could provide complementary information on both the sources and the strength of past bottom-water traffic. The first results on the French Guyana margin suggest the strengthened bottom-water circulation of the glacial analogue of NADW and open up prospects for application in the waters of the Gulf of Guinea.

**Funding:** This research received no external funding.

**Data Availability Statement:** Not applicable.

**Acknowledgments:** This overview represents the accomplishment of a scientific journey of more than 30 years, which has focused mainly in the tropical waters of the Gulf of Guinea. My approach could not have been carried out without the successive collaborations of several colleagues, initially with late Michel Lamboy and Gilles-Serge Odin, and then with late Andrzej Wiewióra and his collaborators, Andrzej Wilamowski and Bożena Lącka. Many works were developed within the framework of the preparation of the doctoral theses of my African students, particularly Congolese, Gabonese, and Cameroonian PhD students, to whom I pay tribute. This text has benefited from the comments and useful advice of three anonymous reviewers.

**Conflicts of Interest:** The authors declare no conflict of interest.

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
