# Peer review of "Quaternary Glauconitization on Gulf of Guinea, Glauconite Factory: Overview of and New Data on Tropical Atlantic Continental Shelves and Deep Slopes"

_minerals, doi:10.3390/min12070908_

Round 1

Reviewer 1 Report

You may cite the following publicationTathagata Roy Choudhury, Sonal Khanolkar, Santanu Banerjee,

Glauconite authigenesis during the warm climatic events of Paleogene: Case studies from shallow marine sections of Western India Global and Planetary Changelume 214

2022

103857

ISSN 0921-8181

https://doi.org/10.1016/j.gloplacha.2022.103857.,,,,,,

This publication relates the authigenesis of glauconite to hyperthermal events.

Author Response

The suggested reference was introduced

Reviewer 2 Report

Comments on MS No. minerals-1792875-peer-review-v1

Quaternary tropical processes of glauconitization on continental shelves and deep oceanic slopes, process comparison. Overview and new data.

1- The text is too long and needs to be shortened and more concise.

2- As a review or overview article it does not provide sufficient data on the glauconite geochemistry.

3- Microscopic petrography with photomicrographs is not included in the article to show morphology, texture, internal structure and clastic impurities of the glauconite grains.

4- There are some unrealistic statements and contradictions in the text. See p. 14 for example where the author states that anoxic conditions tend to transform Fe2+ to Fe3+ and an opposite statement in p. 15 (see annotated text).

 5- Many statements in the text are not supported by references (see my remarks on the annotated text).

6- The conclusions should be presented in a separate section at the end and must include new ideas and new results only and not what previous workers have achieved.

7- Out of 37 references cited in the text there are 16 or more references by the author!

Need to widen the literature review to cover all aspects related to the subject and achieve the target of an overview paper.

8- I suggest reconsidering the keywords and remove the geographic names.

9- The article requires English editing.

10- I suggest major revision of this article.

Author Response

Comments on MS No. minerals-1792875-peer-review-v1

Quaternary tropical processes of glauconitization on continental shelves and deep oceanic slopes, process comparison. Overview and new data.

1- The text is too long and needs to be shortened and more concise.

The characteristic of an overview is to be more extensive than an usual analytical paper. In this context, we do not believe that we have abused this relative freedom. And besides, we don't know how to "shorten" this document and, at the same time « widen the literature review to cover all aspects related to the subject », cf reply to point 7.

The purpose is not to cover all aspects related to the subject and the subject is not glaucontization as a “whole”. This is a synhesis restricted mainly to one region and to late-Quaternary deposits. It turns out that this region, these deposits constitute an exceptional modern "glauconia factory" and that it has been studied extensively by the author.

2- As a review or overview article it does not provide sufficient data on the glauconite geochemistry.

The primary objective of this study is not to present a state of knowledge across the oceans of the world, it is even less intended to consider the many examples of fossil glauconite where the processes of Late diagenesis are particularly taken into consideration. Glauconitization on the shelves of the Gulf of Guinea (and the margins of Ghana and Guyana) is exceptionally well documented. These zones provide access to a resolution rarely achieved elsewhere, both in the field of very high resolution of observations and in that of the chronometry of processes. The geochemical data are presented and commented on each time they could improve our knowledge in the context of the Gulf of Guinea

The deliberate choice was made to present only the most decisive information and, if possible , rather new.

Several paragraphs are thus concerned with geochemistry :

  1. Faecal pellets glauconitization on the Gulf of Guinea shelf

7.1 Ivory Coast-Ghana continental margin

7.2 Demerara margin (French Guiana)

  1. Schematization of cation exchanges during the pellets glauconitization

9.3 Iron determinant intervention and Pleistocene timing of glauconitization

9.5 Neodymium isotopes in glauconite as a tool for palaeoceanographic reconstruction

As this study does not aim to provide an exhaustive state of our current knowledge on glauconite (as reviewer seems envisage …), the title has been modified and the intention of the study proposed in the introduction has been better delimited in order to reduce ambiguity.

New title:

Quaternary glauconitization on Gulf of Guinea, glauconite factory. Overwiew and new data of tropical Atlantic continental shelves and deep slopes.

New introduction entry:

This study does not pretend to propose a synthetic work which would consider the exhaustive state of our knowledge on glauconitization through the oceans of the world. It aims even less to deal with fossil glauconite discovered in Cenozoic or even older terrains. Its primary objective is to present a review of knowledge on the recent processes of glauconitization in the southern Gulf of Guinea (Congo, Gabon, Ghana, in particular). This sector is undoubtedly one of the most efficient ocean "factories" of glauconite and, as such, should soon be the target of new and important research programs. The data that we will present here are the result of a scientific journey of some 30 to 40 years in which the author has had the opportunity several times to participate. Alongside several fairly recent articles, some older ones are little known because they are partly written in French and published in journals with limited circulation. This present writing will be an opportunity to make them public after having revisited them and extracted the most essential points in the light of last scientific progress.

3- Microscopic petrography with photomicrographs is not included in the article to show morphology, texture, internal structure and clastic impurities of the glauconite grains.

The choice was made to present here in the first place original and little distributed documents. But in order not to be too long and to avoid banality, it was not considered necessary to show photos published in numerous publications, which are now very well known to everyone and which would not bring anything specific or new (I think in particular of surface cracks).

Here again, the satisfaction of this request would go against the effort of conciseness which is recommended to us elsewhere.

4- There are some unrealistic statements and contradictions in the text. See p. 14 for example where the author states that anoxic conditions tend to transform Fe2+ to Fe3+ and an opposite statement in p. 15 (see annotated text).

It must be understood that in a first phase, Fe2+ is soluble and mobile in the anoxic environment of the mud, in a second phase, a large part of this iron is concentrated in the microenvironment of the pellet where it oxidizes and attaches to the state of Fe3+. The sentence was misread, that means it was badly written, consequently, it was rewritten.

. Fe2O3 ferrous or ferric iron ? Fe2O3 contents means ferric iron calculation… All iron contents are expressed as Fe3+, even if everybody knows that there is allways some Fe2+ located in octahedral position..

In Guinea Gulf, Fe2+ and Fe3+ discriminations were obtained with Mossbauer methods. Several measurements (26, 27) have shown in the Gulf of Guinea an Fe2+/Fe3+ ratio of between 0.08 and 0.24, a ratio which increases in connection with the increasing proportion of associated goethite. The highest Fe3+ contents are found in the shallowest sediments where goethite and quartz grains are more abundant. These Mossbauer references are therefore added to the text, even if they are works of the author … sorry.

Giresse, P., Wiewiora, A., Lacka, B., 1987. Migration des éléments et minéralogenèse dans les grains verts récents au large de l’embouchure du Congo. Archiwum Mineral., 42, 5-30.

Giresse, P., Wiewiora, A., Lacka, B., 1988. Mineral phases and processes within green peloids from two recent deposits near the Congo River mouth. Clay Mineral, 23, 447-458.

Use of Mossbauer method is linked to a cristallographic prospect of tetahedral and octahedral structure, but it is not a usual routine measurement. Here, on the statistic basis of EDAX microprobe analysis, this valence discrimination cannot be calculated every time.

. more than 3 wt.% is right (Fig. 2b), so not 0.3 wt.% …

5- Many statements in the text are not supported by references (see my remarks on the annotated text).

This remark is mainly addressed to the Introduction for which we have chosen to be concise, leaving to later, in the following paragraphs, the opportunity for a more precise presentation and therefore more exhaustive references. Moreover, as indicated, this writing does not intend to present a global overview, its intention is more regional. However, these references have been introduced or extended whenever indicated by the reviewer.

. 6 2.1 … Atlantic margin… OK

. The potassium content of the glauconite is another indicator of residence time (Odin and Matter, 1981 ; Odin and Fullugar, 1988 ; Amorosi, 1995) OK references introduced

. 3. Faecal pellets glauconitization on the Gulf of Guinea shelf, Juan Jiménez-Millan introduced even if this process belong to Mesozoic deposition and, probably late diagenetic glauconitization

. you need to mention other factors such as redox conditions availabilty of an Fe-rich precursor this paragraph is dedicated only to the restrictive factor of the geometry of the support, the redox conditions are commented in the paragraph 8. Schematization of cation exchanges during the pellets glauconitization ; availablity of Fe-rich precursor is considered in 2.2. Example of organic and ferruginous accumulations off the mouth of the Congo River and commented in 9.3. Iron determinant intervention and Pleistocene timing of glauconitization

. 6. Proposal for glauconitization chronometry in the Gulf of Guinea OK, references introduced

. 7.1 Ivory Coast-Ghana continental margin references added

. Demerara Margin …AMOC … references added

. Schematization of cations exchange, constitutes a summary chapter of the previous analytical chapters and where all the references are indicated. It was not considered necessary to provide these references a second time. But taking into account reviewer remaks, several references have been introduced in this second version…

6- The conclusions should be presented in a separate section at the end and must include new ideas and new results only and not what previous workers have achieved.

New ideas: yes certainly, new results: it is not especially or necessarily what one expects from an overview…

not what previous workers have achieved: I would point out, in all modesty, that I am one of of these previous workers and therefore quite well placed to comment on them.

In the spirit of this study, I believe that the development of the 5 paragraphs of the chapter Discussion, Conclusions was precisely intended to highlight new and original ideas and results.

This form of writing seems to me to be well suited for an overview form.

But at the request of the reviewer, a conclusion has been added at the end of the text.

7- Out of 37 references cited in the text there are 16 or more references by the author!

Need to widen the literature review to cover all aspects related to the subject and achieve the target of an overview paper.

If the intention of this paper had been to provide a synthesis of the state of knowledge on recent and ancient glauconitizations throughout the world, it is certain that this ratio (16/37) would have been exaggerated and, may be, culpable.

But the theme of this study is regional, it wants to take stock of an important intertropical region in terms of education in the field of recent glauconitogenesis. It does not intend to treat the subject on an universal level. The number of authors who have worked both on this region and on this subject is quite limited, but none have been forgotten. It turns out that the author of the study has worked in particular in the Gulf of Guinea for more than thirty years is logically referenced several times. If our intention had been to analyze the global knowledge on glauconitization, it is likely that the proportion of author citations would have been much more modest. Finally, several of the author's works have been published in French, sometimes at remote dates and in journals that are difficult to access. This Sp issue, with the agreement of the publisher, offered a good opportunity to distribute and to update the main lessons.

Accordingly, and once more, this paper does not intend to provide an exhaustive overview of the subject of glauconitization. It cannot therefore be considered in this context “to widen the literature review and to cover all aspects”. Moreover, to widen the literature review would have the corollary of extending its length, in apparent contradiction with the advice in point 1: The text is too long and needs to be shortened…

However, wanting to take into account this concern of the reviewer, we have completed and clarified the intention of the work in the introduction to better define the ambitions, but also the limits of the study.

8- I suggest reconsidering the keywords and remove the geographic names.

This request is quite revealing of the reviewer's approach. Our study is mainly located in the geographical frameworks of the Gulf of Guinea and the Guyanese margin and this on the scale of the last ten millennia. It cannot be universal, neither on a geographical scale, nor on a chronological scale. Consequently here, the geographical locations all have their full meaning.

9- The article requires English editing.

This is also my a priori opinion. However, the linguistic corrections proposed by the reviewer (and respected by the author) were few. The other reviewers offer only minor corrections.

Anyway, it is certain that verification is necessary in this matter. Scientific editor Dale Du confirmed to me that this would be done by Minerals on the final form of the text.

Reviewer 3 Report

Research paper  has merit for publication. M/s is lucid and good contribution

Author Response

Thanks. No comment.

Round 2

Reviewer 2 Report

In the revised version some some useful changes have been made and the objectives of the paper are better clarified. However, to me as a reviewer and scientific reader, I still find the text too long and needs shortening and reorganization to be concise and easy to follow. Objections by the author on some of my  comments are not justified, such as considering petrography and geochemistry in the comparison of the two glauconite sedimentological settings. I think this MS can be improved if rewritten and better organized.